# The Role of Lymphadenectomy in the Surgical Treatment of Hepatocellular Carcinoma: A Systematic Review and Meta-Analysis

**DOI:** 10.3390/cancers16244166

**Published:** 2024-12-13

**Authors:** Gabriele Spoletini, Alberto Mauro, Miriam Caimano, Giuseppe Marrone, Francesco Frongillo, Salvatore Agnes, Quirino Lai, Giuseppe Bianco

**Affiliations:** 1General Surgery and Liver Transplantation, Fondazione Policlinico Universitario Agostino Gemelli IRCCS, 00168 Rome, Italy; miriam.caimano@guest.policlinicogemelli.it (M.C.); giuseppe.marrone@policlinicogemelli.it (G.M.); francesco.frongillo@policlinicogemelli.it (F.F.); salvatore.agnes@policlinicogemelli.it (S.A.); giuseppe.bianco@policlinicogemelli.it (G.B.); 2Policlinico Umberto I, Sapienza University of Rome, 00161 Rome, Italy; quirino.lai@uniroma1.it

**Keywords:** hepatocellular carcinoma, hepatic resection, lymph node dissection, lymph node metastasis

## Abstract

The role of lymphadenectomy, in addition to hepatic resection, remains controversial in the treatment of hepatocellular carcinoma. We systematically reviewed the relevant literature comparing lymph node dissection combined with HR and with no lymph node removal. Having included more than 40,000 patients, we can confirm a higher mortality rate in patients with lymph node metastases, even after lymphadenectomy, thus not supporting its routine adoption as part of standard liver resection for HCC.

## 1. Introduction

Hepatocellular carcinoma (HCC), which covers up to 80% of liver cancer diagnoses globally, is an aggressive tumor with a poor prognosis, especially in the presence of lymph node metastases (LNMs) [1,2]. Hepatectomy—in the form of resection or liver transplantation—can cure HCC with 5-year survival beyond 70% [3]. However, when LNMs are encountered, survival rates drop below 50% [4]. Besides hematogenous dissemination, lymphatic spread is responsible for tumor recurrence [5]. The incidence of LNM in HCC is known to be lower compared to hilar and intrahepatic cholangiocarcinoma, for which lymphadenectomy is recommended as part of the standard surgical treatment [2]. Nevertheless, for autopsies, the incidence of HCC LNM, identified mainly in regional lymph nodes (LNs), ranges between 23.5% in cirrhotics and 34.8% in non-cirrhotics [6,7]. Despite this concept receiving growing attention, lymph node sampling and/or dissection at the time of surgery remains uncommon and is not supported by guidelines [8,9]. On preoperative imaging and at intraoperative assessment, the suspicion of LN involvement may prompt selective sampling to address the patient for appropriate treatment, often ruling out surgery in the case of LN positivity [10]. In addition, liver inflammation in cirrhotic patients can cause LN swelling, posing a challenge in the preoperative detection of metastatic LNs [10]. In some case series, the combination of hepatectomy and lymphadenectomy in clinically positive LNs was followed by poor survival due to disease recurrence [11]. On the contrary, unexpected intraoperative findings of positive LNs may lead to a shift from one treatment to another, such as redirecting from LT to systemic chemotherapy [4]. Differently from other gastrointestinal malignancies, the value of lymphadenectomy in HCC remains unclear. While its role in staging is evident, the oncologic benefit has not been proven. Whether lymphadenectomy should be recommended to improve patient prognosis is not supported by current evidence. Only a few studies have investigated the role of LND as a tool to improve prognosis in HCC patients undergoing parenchymal resection; nevertheless, they were characterized by heterogeneous patient selection and outcome data.

In this study, we sought to systematically review and analyze all the available evidence around the usefulness of lymph node sampling and/or dissection and their potential effect on survival and post-operative outcomes in patients with HCC.

## 2. Materials and Methods

This review was based on the guidelines for the Preferred Reporting Items for Systematic Reviews and Meta-Analyses (PRISMAs) [12]. The systematic review protocol was registered at PROSPERO (CRD42023409182) and available at https://www.crd.york.ac.uk/prospero/display_record.php?ID=CRD42023409182 (accessed on 11 December 2024).

### 2.1. Information Sources, Search, and Study Selection

A systematic literature search was conducted in March 2023 through online databases, registers, and repositories, using “HCC” AND “lymphadenectomy” as a keyword. MEDLINE (Pubmed), the Cochrane Central Register of Controlled Trials (CENTRALs), ClinicalTrials.gov, Google Scholar, Ovid, Scopus, and DOAJ were investigated. Google Scholar was consulted for the first 10 pages. In addition, the reference list of included studies was examined to find other relevant publications on the topic. No geographic, language, year of publication, or follow-up restrictions were applied. Where a study featured multiple eligible articles, we chose the most recent paper with the most significant number of participants and the most extended duration of follow-up for post-hepatic resection recorded outcomes. Case series, case reports, literature reviews, or studies without adequate comparative analyses were excluded. Screening by title and abstract was conducted by two authors independently. For the eligibility of articles after screening, full texts were retrieved and reviewed. Discrepancies were resolved by consensus or by referring to a third author. We used the automation tool Rayyan to screen articles [13]. The selection process was recorded in sufficient detail to complete a PRISMA flow diagram (Figure 1). The MEDLINE (PubMed) search strategy is given as follows: *((((Lymph Node Excision[MeSH Terms]) OR (“Lymph Node Excision*”[Title/Abstract])) OR (“Lymph Node Dissec-tion*”[Title/Abstract])) OR (lymphadenectom*[Title/Abstract])) AND ((((Carcinoma, Hepatocellular[MeSH Terms]) OR (“Carcinoma, Hepatocellular”[Title/Abstract])) OR (hepatocarcinoma[Title/Abstract])) OR (hcc[Title/Abstract]).*

### 2.2. Eligibility Criteria

We structured the eligibility of included studies for this review of intervention using the PICO (Population, Intervention, Comparison, and Outcome) framework.

Population: randomized controlled trials and comparative or cohort studies of adult (age ≥18 years) patients with HCC undergoing hepatic resection. Publications relating to patients undergoing liver transplantation, including living donor liver transplantation and split-livers, were excluded;Intervention: lymphadenectomy, lymph node dissection, or sampling, combined with hepatic resection;Comparison: no lymph node removal;Outcome: overall survival (OS).

### 2.3. Data Extraction and Risk of Bias Assessment

We used a data extraction form for study characteristics and outcome data, which was piloted in a couple of studies in the review. Data extraction was conducted by two authors. The following data were extracted from each included study:Study characteristics: study authors, year, country, study design, and duration of follow-up;The intervention/s of interest: type of intervention, who provided the intervention, and the comparator/s used;Characteristics of the study participants: number of participants, age, gender, and any comorbidity;Survival data.

The risk of bias of studies included in meta-analysis was assessed using the Cochrane Risk of Bias 2 (ROB-2) [14] tool with five assessment domains (randomization process, deviations from intended interventions, missing outcome data, measurement of the outcome, and selection of the reported result) and the Risk of Bias in Non-Randomized Studies of Interventions (ROBINS-I) [15] tool with seven assessment domains (confounding, the selection of participants, classification of interventions, deviation from intended interventions, missing data, measurement of the outcomes, and selection of the reported results). We evaluated the risk of bias for each domain as ‘low’, ‘some concern’ or ‘high’ with the ROB-2 tool and ‘low’, ‘moderate’, ‘serious’ or ‘critical’ with the ROBINS-I tool based on the information presented in each study. Two authors independently assessed the risk of bias for each study, and the interrater agreement using the weighted Kappa and percent agreement was evaluated. Any discrepancies were resolved by consensus or by a third reviewer. Robvis web app was used for visualizing risk of bias assessments [14].

### 2.4. Analysis

A qualitative summary of the results, using text and tables, was performed. RevMan 5 was used to calculate the treatment effect. The effect size for dichotomous outcomes was expressed as the Odds Ratio (OR) with a 95% confidence interval (CI) and hazard ratio (HR) for time-to-event outcomes with standard errors (SEs). A *p*-value < 0.05 (two-tailed) was considered to indicate statistical significance. The HR of the 5-year OS was derived using formulas from Parmer’s article [16]. We planned the meta-analysis to see if there were 2 or more studies with the same outcome. Heterogeneity was assessed using the χ^2^ and I^2^ statistic. If I^2^ > 50%, a random-effect meta-analysis was performed; otherwise, a fixed-effect model was utilized. We assessed publication bias using Egger’s test and performed a funnel plot analysis using R software version n. 4.3.3 if there were 3 or more studies per meta-analysis.

## 3. Results

### 3.1. Study Search Results

A systematic literature search was conducted in March 2023. After the deduplication process, we screened 292 records by title and abstract. In total, 271 were excluded because there was no relevant or since they did not meet the criteria for language or publication type (e.g., editorials, case reports, comments, etc.). The remaining 21 publications were full-text articles and were reviewed for eligibility. Overall, 7 were excluded (5 did not evaluate the effect of lymphadenectomy on survival analysis; 2 had a small sample size—Appendix A). Fourteen studies [17,18,19,20,21,22,23,24,25,26,27,28,29,30], published from 2004 to 2022, that reported the role of lymphadenectomy or LN status for prognostic analysis in a study population of patients with HCC, were included in this review. Among them, only 8 were statistically analyzed [18,19,20,22,24,25,28,29]. The PRISMA flow chart (Figure 1) shows the details of the selective process of this study.

#### Characteristics of Eligible Studies

The included articles were prospective [17,19,20,23] and retrospective [18,21,22,24,26,27,28,29,30] studies. One randomized controlled trial (RCT) [25] was identified. Most studies were conducted in East Asia, while two studies [17,20] were from Europe, and two [28,29] were from the US. The largest ones, with a patient population of 3766 and 14,872, respectively, used the Surveillance, Epidemiology, and End Results (SEER) database, the National Cancer Data Base (NCDB) of the United States, or the Japanese Nationwide Survey. Two registry studies [29,30] collecting data in the SEER database for the same period, despite possibly overlapping populations, were both included since they investigated different outcomes. Among the 39,823 patients diagnosed with HCC, 26,833 were male (67.4%). Minor hepatic resection (<3 segments) was the most commonly performed type of operation (59%). Five studies classified tumor malignancy according to the most up-to-date AJCC staging system. Among the studies reporting pTNM data, most patients had pT1-2 stages. Among all the studies included, 13,898 patients underwent LND (including those who had a lymph node biopsy due to surgical/anatomical difficulties 18.22). The pN stage was reported only for patients with a lymphadenectomy (LNM = 583—4.2%). The incidence of LNM in the included studies was variable due to the intent of LND and the study design: selective LND was performed for preoperatively diagnosed or intraoperatively suspected LNM [18,21,22,23,24,26,30]; otherwise, an unselective LND [17,20,25] was carried out. No additional information was reported in the remaining cases. Metastatic lymph nodes were found in regional and non-regional sites (common hepatic artery, hepatic hilum, para-aortic area, pericholedochal area; peripancreatic head, hepatoduodenal ligament, retro-pancreatic space, hepatic pedicle, celiac trunk, root of mesentery, left gastric artery, abdominal aorta, cervical area). The characteristics of included studies and clinicopathological features of patients are synthetized and presented in Table 1 and Table 2, respectively.

### 3.2. Risk of Bias Assessment

The quality assessment of the included studies in meta-analyses was performed by ROBINS-I and by ROB2 based on the study design. The results of all assessment domains are depicted in Figure 2. All studies showed a low risk of bias for the classification of interventions and measurement of outcome domains. However, registry-based reports did not provide enough information on the deviations from intended interventions. Most articles missed information regarding baseline confounding, thus limiting the reliability of the results above all regarding the investigation of prognostic features of lymph node dissection on survival outcome. The only RCT was judged as low risk of bias across all domains.

### 3.3. Analysis Results

A total of 27,192 participants were selected from eight studies that reported the association between LNM and survival outcomes and the effect of lymphadenectomy in HCC patients [18,19,20,22,24,25,28,29]. Four studies [18,22,24,28] were used to investigate the value of the proven metastatic LN status (LNM) on patient survival. The control group was composed of patients with clinically negative LNs, receiving no LND. The association of LN status with mortality rate was analyzed using fixed-effects model meta-analyses. At 1 (OR 3.25, 95% CI 2.52–4.21; *p* < 0.001), 3 (OR 3.79, 95% CI 2.74–5.24; *p* < 0.001), and 5 years (OR 3.92, 95% CI 2.61–5.88; *p* < 0.001), LNM patients had higher mortality rates compared to the no-LND patient group (Figure 3). There was no significant heterogeneity in these meta-analyses (Figure 3).

To further investigate the predictive value of LNM status on patient survival, the meta-analyses of three studies were carried out, comparing two groups of patients undergoing LND and, therefore, with histologically proven LN status [19,28,29]. A group of patients with LNM was compared with a group with negative LN (LN0). Whereas at 1- year there was no significant difference between the two groups (OR 1.75, 95% CI 1.0–3.04; *p* = 0.05), at 3- (OR 2.88, 95% CI 1.79–4.63; *p* < 0.001) and 5-years (OR 2.54, 95% CI 1.33–4.84; *p* < 0.001) the mortality rate in the LNM group was higher than in the LN0 group (Figure 4). Between-study heterogeneity was moderate through all three meta-analyses (I^2^ > 50) (Figure 4); therefore, the random-effect method was used. Further analysis to investigate the effect of other variables was not possible due to insufficient data from the original articles.

Even though publication bias was performed on a small number of studies (<10), all studies were evenly distributed on the graphs (Appendix A), indicating that meta-analyses were without publication bias (*p* > 0.05).

Finally, only two studies evaluated if regional LND could be performed routinely during liver resection for HCC [20,25]. The comparative analysis to assess the effect of the intervention on overall survival was, therefore, planned on a patient population with unsuspected LNs (i.e., without any known preoperative or intraoperative LNM). There was no significant difference between the two groups (HR 0.93, 95% CI 0.86–1.01; *p* = 0.09), and between-study heterogeneity was not present (χ^2^ = 0.76, df 2, *p* = 0.68; I^2^ = 0%) (Figure 5), arguing that the LND brings no benefit to OS (Figure 5).

## 4. Discussion

The clinical practice of LND during surgery for HCC is extremely heterogeneous, ranging between 10% and 30%, according to the literature [18,22,30]. Instead, out of almost forty thousand patients included in this study, approximately one-third underwent LND (or at least LNS). The inherent selection bias introduced by the focus of our study certainly overestimates the number of patients undergoing LND. However, great variability also derives from geographical differences, with Eastern centers performing LND more commonly than Western ones [30,31]. In general, LND is routine during the curative-intent surgical resection of several gastrointestinal malignancies such as gastric [32], pancreatic [33], and gallbladder cancers [34], with gastric cancer probably having the most standardized approach in this regard [32]. Nevertheless, the role of LND in HCC keeps gaining interest, as evidenced by the recent release of the first RCT comparing LND against no LND [25].

In Western countries, lymphadenectomy is often approached more selectively, guided by clinical and pathological factors to minimize the risk of complications and preserve quality of life. Whether LND confers survival benefits and better staging and at what cost in terms of morbidity is not clear yet. Some authors consider LND an important step during curative-intent surgery for HCC in order to appropriately stage and guide perioperative management [19,22,30,31]. Conversely, other studies have failed to identify the prognostic role of LN status on survival and, thus, have recommended against routine regional lymphadenectomy due to the increased morbidity [10,20].

Lymphadenectomy is generally considered a safe procedure, though it might increase the risk of perioperative complications [10,17,25,30]. Patient and surgical factors, such as cirrhosis status and the extent of the hepatectomy, require careful consideration when evaluating the patient for lymphadenectomy, as they may influence outcomes [35,36]. For patients who underwent hepatectomy combined with LND, liver-specific complications (e.g., post-hepatectomy liver failure, hemorrhage, bile leakage) were more severe and common in HCC patients compared to other liver tumors [17,36].

To date, our study has assembled the largest patient population, analyzing the role of LND in HCC patients.

As expected, our data confirm the worse prognosis of patients with preoperatively suspected positive LN and of those with LNM who underwent LND.

Our analysis was focused on three main aspects. Firstly, comparing 248 patients with LNM at LND (four studies included) with a control group of 24,075 patients with clinically negative LNs (thus not receiving LND); we registered higher 1- and 5-year mortality rates in the former group compared to the latter. This is in line with the current knowledge, as patients in the control group are expected to have better prognosis thanks to the supposed lack of LN metastasis as per preoperative imaging. How LND affected survival by potentially increasing the risk of morbidity is a datum that could not be extrapolated from the included studies due to insufficient data and heterogeneous reporting. Nevertheless, in a mixed population of patients undergoing LND for liver metastases and HCC, Ravaioli et al. reported comparable morbidity between patients undergoing LND and those who did not, except for higher rates of ascites in LND patients [20]. Despite not reporting increased morbidity, Sun et al. recommend routine exploration rather than upfront LND (i.e., LND to be performed, only if surgical exploration turns positive) in cirrhotic patients due to the potential increased risk of complications and the lack of evident curative advantage [18].

Altogether, the reported LND-related morbidity in the studies ranged between 15%, 18%, and 43% [20], with ascites and liver failure as the main complications. As such, some authors consider liver resection combined with lymphadenectomy as a safe procedure that can be performed in all patients in order to determine the precise stage of the disease [19]. Others [18,20,24,28] agree that LND should be performed only on a specific subgroup of patients (those more at risk of harboring LNM); as a matter of fact, the augmented risk of developing intra- and post-operative compilations is considered as not worthwhile due to the lack of evidence in providing improved OS.

Secondly, the meta-analysis of the studies comparing patients who all underwent LND (three studies, 2800 patients included) showed no difference in 1-year overall survival between LNM and LN0 patients. Instead, the difference became evident between 3 and 5 years after surgery, with the LN0 patients surviving for longer, as expected. This might reflect comparable post-operative outcomes, regardless of LND, whilst HCC tumor behavior likely plays a role in the long term once surgical variables become less important [37,38,39]. Nevertheless, performing LND does not provide a survival advantage for those patients who turn out to have LNM, as their overall survival is substantially shorter than those of LN0 patients. Also, a higher incidence of LNM was found in patients with multiple nodules and vascular invasion, reflecting the more advanced tumor stage found by Xiaohong et al. [19] and, therefore, lower chances of cure. Similarly, Bergquist et al. [29] and Kemp Bohann et al. [28], respectively, in their SEER and National Cancer Data Base registry analysis studies, found more advanced tumor histologic features in patients with LNM at lymph node surgical sampling.

As per the third meta-analysis, when LND is performed regardless of the preoperative suspicion of LN positivity, no survival benefit was encountered. It can be noted that both studies adopted a “preventive lymphadenectomy” methodology, with all patients enrolled having no preoperative suspicion of LNM and, eventually, only one patient having a single LNM on the final histopathology report [20,25].

Our study has some limitations. One is the small number of studies included, although several new studies, including a randomized controlled trial, have been published since the previous meta-analysis on this topic [37]. In addition, the studies extracted are mostly based on either retrospective or registry data, which inevitably harbor selection bias and missing data. The effect of LND on survival outcomes remains hard to assess due to the various approaches adopted for the management of HCC. Additional treatments to surgical resection, as well as differences in the baseline of patients, may not have been taken into account in the analyses and, therefore, may have influenced the interpretation of post-operative events. Studies included in the review showed high heterogeneity in terms of the definition of lymphadenectomy (as summarized in Appendix A), indications for surgery, and outcomes. The extent of LND, the minimum number of LNs removed, and the indication for LND are center-specific and often relate to the surgeon’s experience and intraoperative evaluation. LN status for the LN-negative patient population, used as the control group, was defined by the histological examination of dissected LNs [18,19,22,24,28] or because LNs were unsuspected of preoperative imaging findings [22,25], which can also be open to misinterpretation and heterogeneity across the studies.

In our study, we based our analysis on the already existing evidence in the literature, according to which the rate of pathologically positive nodes among clinically nonsuspicious nodes is extremely low (around 1% [10]). This assumption was also confirmed by Ravaioli et al. in their perspective study [20].

It should also be considered that our study population consists of patients affected by surgically resectable disease, which is a liver cancer with intrinsic characteristics (diameter, number of lesions, localization), making it less likely to have already spread to the lymph nodes. We relied on the aforementioned data from the literature and on the features of patients included in our study design. On the contrary, patients with a higher pTNM HCC are undoubtedly more prone to developing lymph node metastases, but they were not included in our study population due to being considered unresectable. 

Another limitation of our study relates to the approach adopted for analyzing survival outcomes. We opted for a meta-analysis using specific time-point survival rates (1-, 3-, and 5-year) as endpoints instead of time-to-event data due to the unavailability of hazard ratios (HRs) for overall survival in the included studies and the limitations related to proportional hazard assumptions. Although HR is critical for understanding risks over time, we chose to utilize ORs at specific time-points to provide a more uniform and unbiased analysis based on the available evidence. Considering methodological decisions and limitations, it is essential to acknowledge the inherent weaknesses/limitations of our study. Although efforts were made to address these limitations through careful study design and reliance on robust existing literature, our data should be considered with caution in light of these factors. Future studies, based on prospective data with standardized lymphadenectomy protocols and comprehensive survival data from time-to-event, could increase the robustness of analyses in this field.

## 5. Conclusions

In conclusion, although regional LNMs have been identified as an unfavorable prognostic factor for HCC, LND does not confer survival benefits in patients with HCC. For this reason, current evidence does not support performing it as part of a surgical routine, whilst its appropriateness may be evaluated in a case-by-case setting.

## Figures and Tables

**Figure 1 cancers-16-04166-f001:**
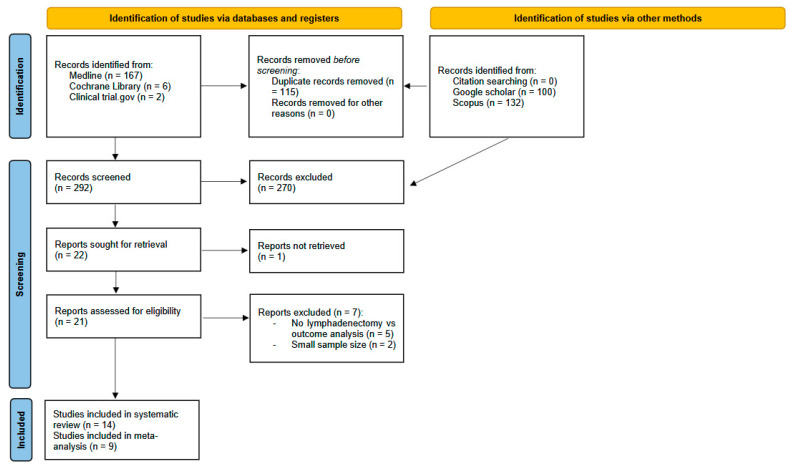
PRISMA 2020 flow diagram. Selection of studies in review of the role of lymphadenectomy in HCC patients.

**Figure 2 cancers-16-04166-f002:**
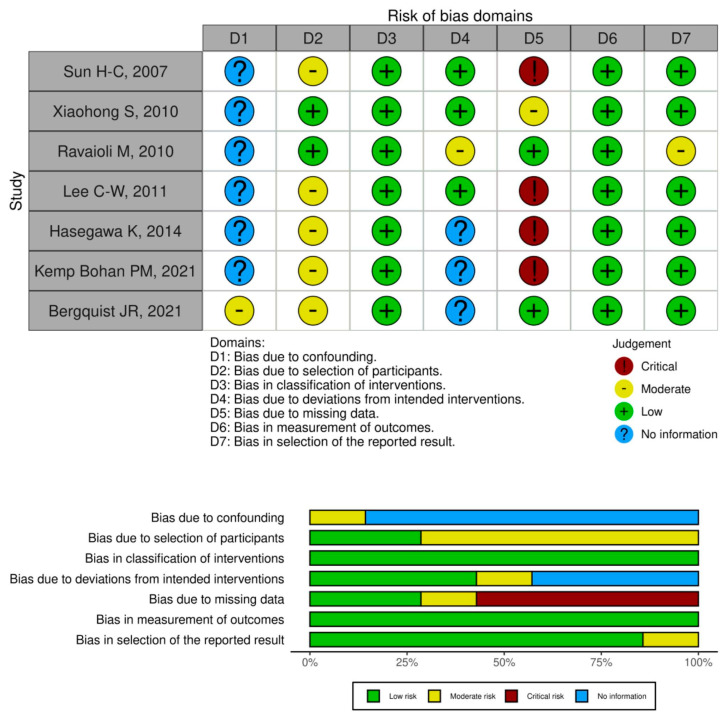
Traffic light (**up**) and domain summary (**down**) plots of risk of bias assessment for OS outcomes from the included studies in the meta-analysis [18,19,20,22,24,28,29].

**Figure 3 cancers-16-04166-f003:**
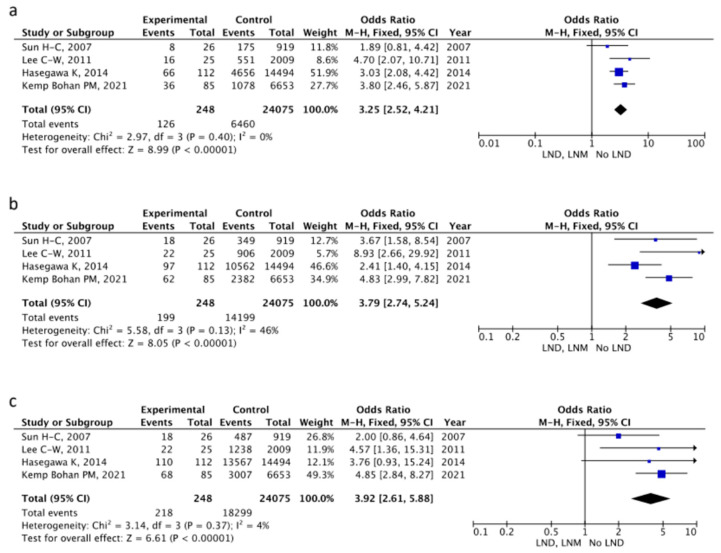
Forest plot comparing (**a**) 1 y, (**b**) 3 y and (**c**) 5 year mortality in LND, LNM vs. no-LND groups [18,22,24,28].

**Figure 4 cancers-16-04166-f004:**
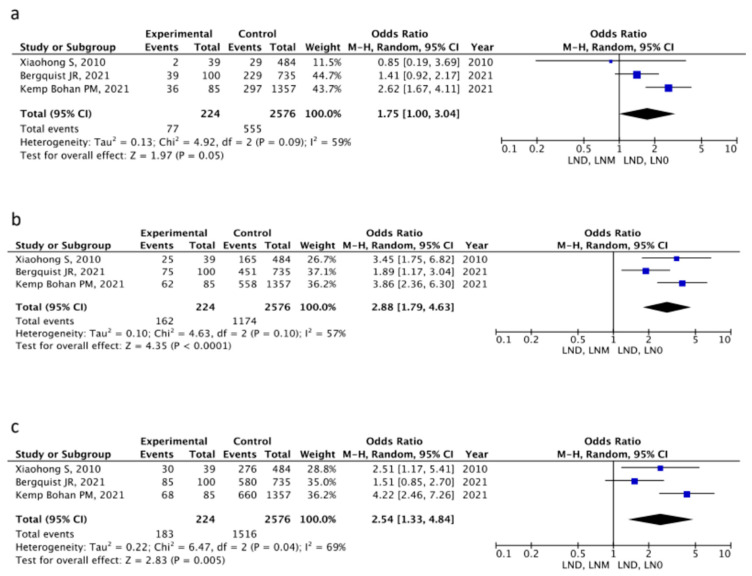
Forest plot comparing (**a**) 1 y, (**b**) 3 y and (**c**) 5 year mortality in LN0 vs. LNM groups [19,28,29].

**Figure 5 cancers-16-04166-f005:**
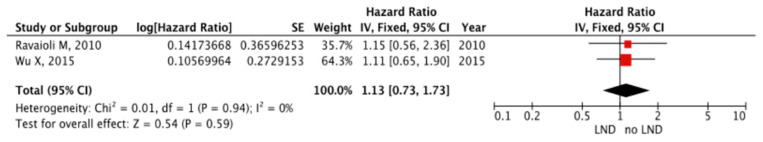
Forest plot comparing overall survival in LND vs. no-LND groups [20,25].

**Table 1 cancers-16-04166-t001:** Characteristics of included studies.

Ref, Year of Publication	Country	Type of Study	Period of Inclusion	Study Population N	Male (%)	Age	Type of Surgical Resection (%)
Ercolani G, 2004 [17]	Italy	Prospective cohort study	1999–2001	40	-	-	Minor resection (87.5) Major resection (12.5)
Sun H-C, 2007 [18]	China	Retrospective case–control study	1999–2005	968	850 (87.8)	50.7 ^a^	Not mentioned
Xiaohong S, 2010 [19]	China	Prospective cohort study	2001–2004	523	453 (86.6)	-	Not mentioned
Ravaioli M, 2010 [20]	Italy	Prospective cohort study	2001–2005	75	-	-	Not mentioned
Kobayashi S, 2011 [21]	Japan	Retrospective case–control study	1992–2008	75	16 (88.9) ^§^	65.2 ^a§^	Minor resection (44.4) Extended lobectomy (5.5) Non-operative treatments (16.7) None (33.3)
Lee C-W, 2011 [22]	China	Retrospective case–control study	1982–2005	2034	1594 (78.33)	55 ^b^	Not mentioned
Awazu M, 2013 [23]	Japan	Prospective cohort study	1998–2012	15	13 (86.7)	63.1 ^a^	Partial hepatectomy (13.3) Hemihepatectomy (33.3) Segmentectomy (6.7) Non-operative treatments (6.7) None (33.3)
Hasegawa K, 2014 [24]	Japan (JNS)	Retrospective nationwide survey study	2000–2005	14,872	11,502 (77.3)	66 ^b^	Not mentioned
Wu X, 2015 [25]	China	Randomized controlled trial	2005–2012	79	65 (82.3)	51.4 ^a^	Minor resection (53.2) Major resection (46.8)
Tomimaru Y, 2015 [26]	Japan	Retrospective cohort study	1980–2012	38	34 (89.5)	61 ^a^	Not mentioned
Yang A, 2019 [27]	China (SEER)	Retrospective population-based study	2004–2013	3766	2672 (71)	60 ^b^	Not mentioned
Kemp Bohan PM, 2021 [28]	US (DCDB)	Retrospective hospital-based study	2004–2015	8095	5780 (71.4)	64 ^b^	Partial Hepatectomy (60.2) Anatomic Hepatectomy (32.2) Extended Hepatectomy (7.5)
Bergquist JR, 2021 [29]	USA (SEER)	Retrospective population-based study	2003–2015	5395	3854 (71.4)	63.5 ^b^	Minor resection (57.4) Major resection (34.1) Extended resection (8.1)
Chen X, 2022 [30]	China (SEER)	Retrospective population-based study	2004–2015	3905	-	-	Not mentioned

^§^ Available data only for patients with LNM histologically confirmed. Mean ^a^, median ^b^. NR, not reported; NA not available; SEER, The Surveillance, Epidemiology, and End Results US registry; NCDB, National Cancer Data Base United US registry; JNS, Japanese Nationwide Survey.

**Table 2 cancers-16-04166-t002:** Clinicopathological features of patients.

Ref	Staging System	p STAGE pT1/T2 n (%)	Regional LND n/N	LNM n (%)	LN Yield Mean (SD)/ [R]	Site of LNM	Additional Treatments	
							LND n	No LND n
Ercolani G [17]	Not mentioned	Not mentioned	40/40	3/40 (7.5)	At least 4	RP CHA HP (periportal)	Not mentioned	Not mentioned
Sun H-C [18]	6th edition of UICC TNM	864 (89.2)	55 */968	49/55 (89.1)	At least 3	HH PPh PAa major omenta PDa	0	23 AT
Xiaohong S [19]	5th edition of AJCC TNM	Not mentioned	523/523	39/523 (7.4)	6.6 (2.7) [3,4,5,6,7,8,9,10,11,12,13,14,15,16,17,18,19,20,21,22,23,24,25]	HP RPs CHA	Not mentioned	Not mentioned
Ravaioli M [20]	Not mentioned	Not mentioned	37/75	1/37 (2.7)	At least 4	Not relevant	NA	NA
Kobayashi S [21]	Liver Cancer Study Group of Japan	8 (44.3)	20/75	18/20 (90)	Not mentioned	LGA PPh AA D HDL CHA	5 NAT + 8 surgeries	55 PC+AT
Lee C-W [22]	6th edition of AJCC TNM	1081 (53.2)	170 */2034	25/170 (14.7)	3.08 [1,2,3,4,5,6,7,8]	CHA HH HDL	Not mentioned	Not mentioned
Awazu M [23]	Not mentioned	5 (33.3)	15/15	15/15 (100)	1.8 (0.4)	HDL CHA RPs PAa CT RM Ca	8 NAT + 4 surgeries 11 AT + 5 surgeries	0
Hasegawa K [24]	Liver Cancer Study Group of Japan	9089 (61.1)	9591/14,872	118/9591 (1.2)	Not mentioned	Not mentioned	Not mentioned	Not mentioned
Wu X [25]	Not mentioned	75 (94.9)	41/79	0/41 (0)	2.8 (1.5)	Not relevant	16 AT + surgery	14 AT + surgery
Tomimaru Y [26]	Not mentioned	Not mentioned	25 */38	25/25 (100)	Not mentioned	PA HH CHA	3 AT	6 AT + 7 none
Yang A [27]	6th edition of AJCC TNM	2995 (79.5)	538/3766	49/538 (9.1)		Not mentioned	Not mentioned	Not mentioned
Kemp Bohan PM [28]	6th edition of AJCC TNM	6355 (78.5)	1442/8095	85/1442 (5.9)	At least 2	Not mentioned	135 NA T 132 A T	319 NA T 424 A T
Bergquist JR [29]	Not mentioned	4335 (80.3)	835/5395	100/835 (12.0)	At least 1	Not mentioned	Not mentioned	Not mentioned
Chen X [30]	8th edition of AJCC TNM	3122 (79.9)	566/3905	56/1346 (4.2)	1.94	Not mentioned	0	146 NA T

NAT, neoadjuvant treatment; PC, palliative care; AT, adjuvant treatment; UICC, The Union for International Cancer Control; AJCC, American Joint Committee on Cancer; SD, standard deviation; R, range. * SUN: 32 LNDs and 23 incisions or aspiration biopsies for unresectable LNM (invasion of major vessels or dorsal area of pancreas, otherwise only incisional or aspiration biopsy); LEE: 170 LND+ biopsies (25 pts with LNM/22 LNDs + 3 biopsies); TOMIMARU: in 22, LN was completely removed whereas in 3, incomplete excisions were made. CHA, common hepatic artery; HP, hepatic pedicle; HH, hepatic hilum; PAa, para-aortic area; PDa, pericholedochal area; PPh, peripancreatic head; HDL, hepatoduodenal ligament; RPs, retro-pancreatic space; CT, celiac trunk; RM, root of mesentery; LGA, left gastric artery; PPh, posterior surface of the pancreas head; AA, abdominal aorta; Ca, cervical area.

## Data Availability

https://www.nlm.nih.gov/index.html (accessed on 17 March 2023); https://www.cochranelibrary.com/central (accessed on 17 March 2023); https://clinicaltrials.gov; https://scholar.google.com (accessed on 17 March 2023); https://ovidsp.ovid.com (accessed on 17 March 2023); https://www.scopus.com/home.uri (accessed on 17 March 2023); https://doaj.org (accessed on 17 March 2023).

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
