# Peer review of "The Role of Lymphadenectomy in the Surgical Treatment of Hepatocellular Carcinoma: A Systematic Review and Meta-Analysis"

_cancers, 2024, doi:10.3390/cancers16244166_

Round 1
Reviewer 1 Report
Comments and Suggestions for Authors
A systematic review and meta-analysis for lymphadenectomy in HCC resection.
Few comments:
1. the keywords used for the literature search are limited with "HCC" and "lymphadenecto*" only. Potentially some studies could be missed.
2. It is unclear to me if only English publications were included. If it is, there could be potential selection bias.
3. The definition of "Lymph Node Dissection (LND)" is varied between studies. This was touched on briefly in the "Limitation" section. However, it might be useful to have a table listing the definition of LND among the studies.
Author Response
Comment 1: "The keywords used for the literature search are limited with "HCC" and "lymphadenecto*" only. Potentially some studies could be missed.
Authors’ response: We developed a search strategy based on a combination of the mentioned key words and their related synonymous and MeSH terms. In addition, we have expanded the study research on reference list of included studies and relevant reviews published on the topic.
The search strategy for PubMed we adopted was:
((((Lymph Node Excision[MeSH Terms]) OR ("Lymph Node Excision*"[Title/Abstract])) OR ("Lymph Node Dissection*"[Title/Abstract])) OR (lymphadenectom*[Title/Abstract])) AND ((((Carcinoma, Hepatocellular[MeSH Terms]) OR ("Carcinoma, Hepatocellular"[Title/Abstract])) OR (hepatocarcinoma[Title/Abstract])) OR (hcc[Title/Abstract]))
Comment 2: "It is unclear to me if only English publications were included. If it is, there could be potential selection bias."
Authors’ response: We did not pose any language restrictions in the research search. In particular, except for two records in our native language, we found 42 published in a foreign language other than English (e.g. Japanese, German, Spanish). None of these records were selected because they were not relevant to our research question.
We have stated the suggested information as follows: “No geographic, language, year of publication or follow-up restrictions were applied.”.
Comment 3: "The definition of "Lymph Node Dissection (LND)" is varied between studies. This was touched on briefly in the "Limitation" section. However, it might be useful to have a table listing the definition of LND among the studies."
Authors’ response: The heterogeneity of the definition of Lymph Node Dissection has been investigated in the “limitation” section and, as suggested by the Reviewer, we have added a table listing the definition of LND across studies.
“Studies included in the review showed high heterogeneity in terms of the definition of lymphadenectomy (as summarized in Table S1 - supplementary), indication for surgery and outcomes. The extent of LND, the minimum number of LNs removed and the indication for LND are center-specific and often related to surgeon experience and intraoperative evaluation. LN status for the LN-negative patient population, used as control group, was defined by histological examination of dissected LNs [18,19,22,24,28] or because LNs were unsuspected at preoperative imaging findings [22,25], which can also be open to misinterpretation and heterogeneity across the studies.”
Reviewer 2 Report
Comments and Suggestions for Authors
This study was aimed to assess the role of lymph nodes dissection for HCC undergoing hepatectomy. This issue has been not clearly investigated previously, and therefore no clear evidence has been obtained yet. From the background of previous studies, this study might be clinically implicative at the present time.
1, It is difficult to suspect how many patients have proven lymph node metastasis in patients who underwent hepatectomy without lymph nodes dissection as clinically negative lymph nodes metastasis. How did authors speculate the number of patients with lymph nodes metastases proven in clinically negative lymph node metastases?
2, What was the result of comparison of overall survival between patients with proven positive lymph nodes metastases and proven negative lymph nodes metastases in the group of patients who underwent lymph nodes dissection with hepatectomy? It should be revealed clearly in the text.
3, What was the result of comparison of overall survival between patients with proven lymph nodes metastases in patients who underwent lymph nodes dissection and patients with clinically negative lymph nodes metastases who did not underwent lymph nodes dissection?
It should be shown in the text also.
4, There were some literatures in the “Reference” without showing published articles such as Ref 6, 7,11 13, etc. And also page and volume in the journal were not shown clearly in several numbers of references.
Author Response
Comment 1: "It is difficult to suspect how many patients have proven lymph node metastasis in patients who underwent hepatectomy without lymph nodes dissection as clinically negative lymph nodes metastasis. How did authors speculate the number of patients with lymph nodes metastases proven in clinically negative lymph node metastases?"
Authors’ response: In our study, we based our analysis considering the already existing evidences in the literature, according to which the rate of pathologically positive nodes among clinically nonsuspicious nodes is extremally low (around 1% - Stephen R. Grobmyer et al.). This assumption has been confirmed also by Ravaioli at al. in their perspective study.
It should also be considered that our study population consists of patients affected by surgically resectable disease, that is a liver cancer with intrinsic characteristics (diameter, number of lesions, localization) that make it less likely to have already spread to lymph nodes. We relied on the aforementioned literature data and on the features of patients included in our study design. On the contrary, patients with a higher pTNM HCC are undoubtedly more prone to developing lymph node metastases, but they are not included in our study population due to being considered unresectable.
Comment 2: "What was the result of comparison of overall survival between patients with proven positive lymph nodes metastases and proven negative lymph nodes metastases in the group of patients who underwent lymph nodes dissection with hepatectomy? It should be revealed clearly in the text."
Authors’ response: To provide a more appropriate and unbiased analysis of patients survival, we chose to calculate only a meta-analysis with an endpoint based on a specific time point (1-, 3-, 5-year), instead of overall survival based on hazard ratios (HRs). Our decision was supported by several reasons, mainly due to the absence of hazard ratios (HRs) for overall survival in each of the included studies, also because the survival data are almost limited to survival rates at specific time points; moreover, we could not verify the proportional hazards assumption across studies. Therefore, although complete time data are not available, we considered the usefulness of ORs in providing comparative information at specific time points, and for a consistent analysis, we ensured that the data comparison was uniform, such as the same time point and the same event definition in the studies included in the meta-analysis.
Comment 3: "What was the result of comparison of overall survival between patients with proven lymph nodes metastases in patients who underwent lymph nodes dissection and patients with clinically negative lymph nodes metastases who did not underwent lymph nodes dissection? It should be shown in the text also."
Authors’ response: please refer to the aforementioned considerations.
Comment 4: "There were some literatures in the “Reference” without showing published articles such as Ref 6, 7,11 13, etc. And also page and volume in the journal were not shown clearly in several numbers of references."
Authors’ response: We have reviewed all the references and improved ref n.°: 3, 5, 6, 7, 11, 13, 23.